# Unique Slow Crack Growth Behavior of Isotactic Polypropylene: The Role of Shear Layer-Spherulites Layer Alternated Structure

**DOI:** 10.3390/polym12112746

**Published:** 2020-11-20

**Authors:** Mingjin Liu, Jiaxu Luo, Jin Chen, Xueqin Gao, Qiang Fu, Jie Zhang

**Affiliations:** State Key Laboratory of Polymer Materials Engineering, College of Polymer Science and Engineering, Sichuan University, Chengdu 610065, China; Mingjin1006@163.com (M.L.); luojiaxuzzz@163.com (J.L.); chenjinsc1996@163.com (J.C.); gaoxueqin@scu.edu.cn (X.G.); qiangfu@scu.edu.cn (Q.F.)

**Keywords:** iPP, alternated structure, slow crack growth

## Abstract

With the development of polymer science, more attention is being paid to the longevity of polymer products. Slow crack growth (SCG), one of the most important factors that reveal the service life of the products, has been investigated widely in the past decades. Here, we manufactured an isotactic polypropylene (iPP) sample with a novel shear layer–spherulites layer alternated structure using multiflow vibration injection molding (MFVIM). However, the effect of the alternated structure on the SCG behavior has never been reported before. Surprisingly, the results showed that the resistivity of polymer to SCG can be enhanced remarkably due to the special alternated structure. Moreover, this sample shows unique slow crack propagation behavior in contrast to the sample with the same thickness of shear layer, presenting multiple microcracks in the spherulites layer, which can explain the reason of the resistivity improvement of polymer to SCG.

## 1. Introduction

Polymer materials possess great potential as tailoring either the chemical or physical structures of polymer in solid state at different scales can satisfy the requirement for a certain application. Controlling microstructures via adjusting processing conditions (temperature, shear rate, etc.) during processing is more efficient and operable in contrast to chemical modification. This is called the classical “structure–property” relationship, which is a guideline for polymer processing [1,2,3]. Hence, it is of vital importance to reveal the relationship between the microstructure and properties for academia and industry.

Polymer products often fall in a brittle behavior after they are long exposed to the service temperature and low stress, containing the formation of a craze at a point of stress concentration and the subsequent propagation and fracture of the materials [4,5,6,7]. This long-term brittle failure, so-called slow crack growth (SCG), determines the practical service life of materials. A large number of studies have been carried out to understand comprehensively this phenomenon and create the ways to improve considerably the polymer resistance to SCG. It has been confirmed in the previous literature that the molecular topological structure and morphologies, involving the content of tie molecules [8], the number of short-chain branches [9], the molecular weight and its distribution [10], and crystalline morphology [11], are closely related to resistance to SCG. For example, Brown studied the effect of molecular weight and branch density on the rate of SCG, and their results showed that a high molecular weight as well as short chain branch can enhance the resistance to SCG significantly [10,12]. Ludwig et al. discovered that polyethylene (PE) with a broad molecular weight distribution has superior long-term mechanical properties [13]. Meanwhile, many researches have proven that the formation of a shish-kebab structure and more perfect crystals can help to enhance the resistance of polymer to SCG to a large extent [11,14,15].

It is well known that injection molding is one of the most important processing technologies for polymers. Nevertheless, the plastic parts prepared by conventional injection molding (CIM) only have a relatively low content of shish-kebab structure in the skin layer in contrast to the amounts of spherulites that constitute the core layer, resulting in the poor mechanical properties of products. To optimize their performance, the issue of increasing the number of shish-kebab structures via imposing an extra strong shear field on polymer melt, which would result in the “coil-stretch” transition of the molecular chains, has been stressed in the past decades [16,17]. A large amount of experiments have been carried out to demonstrate that some modified injection technologies, such as pressure vibration injection molding (PVIM) [18,19], oscillatory packing injection molding (OSIM) [20,21] and push−pull injection molding [22,23,24] could be applied to obtain self-reinforced parts involving a mass of shish-kebab structure. A novel multiflow vibration injection molding (MFVIM) technology based on PVIM has been proposed by our group in recent years, whose mechanism has been described in the previous papers [18,25,26]. We can not only prepare plastic parts including a high content of shish-kebab to induce self-reinforcement effect, but also parts with a shear layer–spherulites layer alternated structure by tuning the processing parameters such as injection pressure and interval time. It has been reported that the Izod impact strength could be remarkably enhanced for the products with the alternated structure [27,28], and it could be further improved by thermal annealing at a suitable temperature for a certain time. The highest value climbed up to 90 KJ/m^2^ for isotactic polypropylene (iPP), while the value of the sample prepared by CIM is lower than 5 KJ/m^2^ [29]. However, the influence of the distribution of shear layer on the long-term mechanical properties has never been investigated.

iPP, one of the most important general polymeric materials, presents excellent performances and relatively low costs. It was chosen to be the material in the current work, and the injection molded parts with a distinctly different hierarchic structure were prepared by using CIM and MFVIM respectively. On the basis of the previous investigation, we try to understand the relationship between the long-term mechanical properties and unique shear layer-spherulites layer alternated structure for the first time. The microstructure and long-term mechanical properties were detected by polarized optical microscopy (POM), differential scanning calorimeter (DSC), scanning electron microscopy (SEM) and self-designed SCG device [11]. The results indicated that such a special morphology could enhance the resistance of iPP to SCG pronouncedly, and the mechanism was also proposed. This work provided a promising and easy way to alleviate the environmental problems to some extent by lengthening the service life of the products prepared by iPP.

## 2. Experimental Section

### 2.1. Materials

IPP (commercial grade T30S) with the density of 0.910 g/cm^3^ and melt flow index (MFI) of 2.90 g/10 min (230 °C, 2.16 kg), was available from Lanzhou Petrochemical Company (Lanzhou, China). TX-10, as the surfactant, was applied to accelerate the process of slow crack growth.

### 2.2. Samples Preparation

In this work, conventional injection molding (CIM) and multiflow vibration injection molding (MFVIM) samples were prepared by adjusting the pressure and interval time during the packing stage, and they were labeled as CIM, V_1_, V_2_ respectively. V_1_ represents the sample with simply increased thickness of shear layer, while V_2_ is the one with unique shear layer-spherulites layer alternated structure. It should be noted that both of them almost have the same thickness of shear layer. The temperature profile from hopper to nozzle was 160, 180, 190, 200, and 200 °C, respectively, and the mold temperature was fixed at 50 °C. Some molding parameters are listed in Table 1.

After the samples were injection molded, dumbbell bars were cut from the same position for all specimens to investigate the long-term mechanical properties, which were 4 mm width and 3 mm thickness. We made a notch in the middle location for all the samples (shown in Figure 1) by a blade. The crack tips were sharp and the initial crack lengths were measured by vernier caliper. It should be noted that we tested three times for each kind of sample to avoid small differences in the initial crack length that may produce different behaviors.

### 2.3. Polarized Optical Microscopy (POM)

Thin slices with thickness of 30 μm were cut from different samples at the same position by a microtome. Then the slices were observed by a DX-1 (Jiang Xi Phoenix Optical Co., Shangrao, China) microscope connected with a Canon 500D digital camera (Canon, Tokyo, Japan), and the observation direction for POM was parallel to transverse direction (shown in Figure 1).

### 2.4. Differential Scanning Calorimeter (DSC)

A DSC (TA Q200) device was used to analyze the thermal behavior of different samples. All measurements were carried out under dry nitrogen atmosphere. Specimens about 3−8 mg were heated from 80 to 200 °C with a heating rate of 10 °C/min. The following equation was utilized for calculating the total crystallinity, $X_{c}$, of each sample:$$X_{c}=\frac{∆H_{m}}{∆H_{m}^{o}}$$
where $∆H_{m}$ represents the measured value of the enthalpy of fusion and $∆H_{m}^{o}$ manifests the fusion enthalpy of completely crystallized iPP. Here, the value of $∆H_{m}^{o}$ was selected as 207 J/g.

### 2.5. Synchrotron Two-Dimensional X-ray Measurements

2D small-angle X-ray scattering (2D-SAXS) were conducted on the BL16B1 beamline in Shanghai Synchrotron Radiation Facility (SSRF), Shanghai, China. The dimensions of the rectangle-shaped beam were 0.5 × 0.8 mm^2^, and the wavelength of light was 0.124 nm. The sample-to-detector distance was 1900 mm for SAXS.

The long period ($L_{p}$) of SAXS results can be calculated by Bragg’s law as follows:$$L_{p}=\frac{2\pi }{q_{max}}$$
where $q_{max}$ is the peak position of the 1D-SAXS intensity profile. The crystallite thickness ($L_{c}$) is calculated as long period multiplied by the crystallinity. The thickness of the amorphous phase ($L_{a}$) is calculated by $L_{p}$ − $L_{c}$.

### 2.6. Scanning Electron Microscopy (SEM)

A JEOL field emission scanning electron microscope (model JSM7500F, Tokyo, Japan) was employed to carefully observe the fracture morphology of different samples after suffering the slow crack growth process. Before observation, the specimens were gold sputtered.

### 2.7. Slow Crack Growth Process (SCG)

The slow crack growth experiment of the various samples was conducted on the self-designed SCG device. 10% TX-10, as the surfactant, was used to accelerate this process and the test temperature was maintained at 50 °C. The initial stress in the experiment was different for sample CIM (5.9 MPa), V_1_ (7.2 MPa) and V_2_ (7.2 MPa) due to the variation of tensile strength among the samples. We used spring scale to correct the real initial stress loading on the samples. Due to the addition of TX-10, the induction period for the crack was considerably shorter than it would be under the normal conditions, but the process of SCG was the same, which could uncover the relationship between the microstructure and resistance to slow crack propagation. For the accuracy of the experiment, at least three specimens were tested to estimate the slow crack growth process for each sample.

## 3. Results and Discussion

### 3.1. Crystalline Structure

The sample prepared by CIM manifests a typical skin–core structure, which is in agreement with the previous papers published before [2], while sample V_1_ and V_2_ show special microstructure observed from Figure 2. It should be noted here that the thickness of shear layer of samples V_1_ and V_2_ is almost the same (about 55%), indicating the same content of shish-kebab for these two samples, because it was speculated that the shear layer consisted fully of shish-kebab structure. More detailed information about the microstructure of all the samples, such as crystalline morphologies, orientation degree, the content of β crystal, have been reported in our previous literature [29]. Here, we can reasonably conclude that manipulated microstructures with different shear layer distributions could be obtained for the parts through MFVIM.

The melting behavior of samples was estimated by calculating the melting temperature and crystallinity. We obtained these two parameters from shear and spherulites layer respectively resulting from the extremely different structure for shish-kebab and spherulites. It should be noted that just selected data was presented here, because the layers containing the same crystalline morphologies showed the similar value for each sample. For example, the value of $T_{m}$ and $X_{c}$ of layer L2 and L4 for sample V_2_ is almost the same. As illustrated in Figure 3, these three samples presented the analogous melting behavior in the same kind of layer regardless of the shear layer or spherulites layer. The melting temperature and crystallinity of different layers are collected in Table 2. Theoretically speaking, the melting point of shish-kebab should be higher than that of spherulites because the former is regarded as a more thermal stable state. It is noticeable that the melting temperature scarcely changes in spite of the completely varied microstructure for shish-kebab and spherulites.

For the sake of obtaining more information on crystalline structure, SAXS experiment was carried out. The selected results were shown in Figure 4 and the long period and the lamellae thickness of samples were collected in Table 3. From the results presented here, we can clearly know that the long period just changes a little for all the samples, regardless of shish-kebab or spherulites. Specifically, the value of $L_{p}$ of shish-kebab for sample V_1_ and V_2_ is 31.67 and 31.81 respectively, which is only slightly higher than that of specimen CIM (28.98 nm). Meanwhile$,\;L_{p}$ of spherulites remains almost the same. Coupling the DSC with SAXS results, the crystalline thickness $L_{c}$ could be calculated directly (shown in Table 3). It shows that the crystalline size of these three samples, regardless of shish-kebab or spherulites, is relatively similar, indicated by almost the same value of crystalline thickness.

As discussed above, it can be well known that the molecular topological and crystalline structure associated with the capacity for resisting slow crack propagation, such as the content of shish-kebab, the distance between the lamellae and the perfection of the lamellae, are completely similar for sample V_1_ and V_2_ regarding shear and spherulites layer. Hence, the difference between these two samples is just the distribution of shear layer or spherulites layer. Next, we would study the role of alternated structure on the slow crack growth behavior to provide a practical application prospects for sample with such a novel structure. So, the results and discussion are focused on samples V_1_ and V_2_ in the later section.

### 3.2. Slow Crack Growth Process

In the plastic parts applications, SCG is one of principal failure modes, which is initiated by a defect or stress concentration, containing the formation of a craze at a point of stress concentration, the subsequent crack propagation and fracture of the materials. The SCG process would not take extremely long time to complete under the suitable conditions for selected samples. The initial stress we selected for sample CIM was 5.9 MPa, and it was 7.2 MPa for both sample V_1_ and V_2_. The reason why we chose two different initial stresses was the obvious difference of tensile strength between sample CIM and the other two specimens. Further, it has been confirmed that the value of yield strength of samples V_1_ and V_2_ was similar [11], so we used the identical initial stress to investigate the evolution of slow crack growth for these two samples.

Figure 5 shows the patterns of the evolution of crack propagation of samples under a low initial stress for different times at 50 °C. Note that there are some dots in this figure, which is induced by some impurities or bubbles. We can clearly observe that the evolution of the slow crack growth for sample CIM is relatively rapid compared to the other two specimens, the sample completely fractures only within 507 min, which indicates the poor capacity of the resistivity of the material to the slow crack propagation behavior. The phenomenon could be explained by the low content of shish-kebab in the sample manufactured by conventional injection molding. Compared with sample CIM, the total fracture time of sample V_1_ climbs from 507 min to 1380 min originating from the high content of shish-kebab structure induced by the extra imposed shear field, which is consistent with the previous work [11]. These two samples underwent a complete fracture process under the experimental conditions used in this work. The difference of fracture process in the initial stage between samples V_1_ and V_2_ is considerably slight as illustrated in Figure 5. Surprisingly, the fracture behavior of sample V_2_ (shown in Figure 5) at the later stage is totally different from V_1_, showing multiple microcracks except for the original notch in this sample, indicated by the presence of stress white region, which is absent in the sample CIM and V_1_. The loading that the initial notch suffered would be shared through the formation of microcracks, leading to the suppression of slow crack growth. The reason why the stress white region formed will be discussed in a later section.

Figure 6 manifests the patterns of crack propagation of sample V_1_ and V_2_ at the later stage to furtherly reveal the different fracture process. It can be well observed that stress white region is distinctly obvious for sample V_2_ when the fracture time is over 24 h, but not for sample V_1_. Sample V_1_ would undergo the complete fracture process under the low initial stress without microcracks formed. Here, it should be noted that the deformation of sample V_2_ along the tensile direction had reached the measuring limit of SCG device after subjected to the low stress for 2400 min, originating from the formation of numerous microcracks at the later stage. The reason why there are not obvious microcracks could be observed at the early stage may be attributed to the long induction period that defects propagate into microcracks under the low stress. We could speculate that the microcracks formed during the slow crack growth process can prominently decrease the stress at the initial crack tip. That is to say, the stress loaded on the sample can be shared with a large number of microcracks besides the initial notch, which enhances the resistivity of this sample to the SCG process to a large extent. It is of practical or scientific significance for the development of polymer science and engineering. Hence, we can have a reasonable vision that the sample with such a novel shear layer–spherulites layer alternated structure could lengthen the service life of the products under some conditions.

Figure 7 quantitatively characterizes the crack growth process by introducing the crack length versus time curves. It could be well observed that sample CIM shows rapid crack propagation, indicated by fracture within a short time. As for the other two samples, they possess similar fracture process in the early fracture stage, but the crack growth behavior is extremely different during the later fracture stage, the crack propagation of sample V_2_ was suppressed showing the independence of time during the later stage, which is consistent with the result of Figure 5 and Figure 6.

In order to get more detailed information about the fracture morphology of sample V_1_ and V_2_ after the SCG process, Figure 8 presents its SEM images, and the observation direction of SEM is the transverse direction as shown in Figure 1. The extremely different morphologies for sample V_1_ and V_2_ can be well observed. The sample V_1_ (shown in Figure 8a) manifested relative smooth surface with the absence of microcracks. It should be noted clearly that the sample V_2_ does not fracture completely after suffered the SCG process for a very long time, and there are many stress white regions throughout the specimen. Corresponding to the POM photograph presented in Figure 2, it can be known from Figure 8b that microcracks formed in the spherulites layer but not the shear layer due to the inferior properties for spherulites. Figure 8(b_1_,b_2_) show the magnification of morphologies at corresponding locations in Figure 8b. A large amount of microcracks can be observed more clearly, which further confirms the existence of microcracks in this sample. Here, the reason why microcracks formed only in the spherulites layer for sample V_2_ will be discussed later to provide a promising way to manufacture robust plastic parts to meet the requirements of industry.

The SEM images A, B and C, shown in Figure 9, represent the morphology of the whole fracture surface for the samples CIM, V_1_ and V_2_ respectively. While the other two columns, such as A_1_ and A_2_, manifest the magnification of morphologies at the corresponding locations marked in the first column. It is obvious that the fracture surface of sample CIM is relatively smooth in the forepart in contrast to the rear part. Each part represents the fracture process at a different time. Brittle fracture (Figure 9A1) showing smooth morphology appears in the early stage of the crack growth. Then plastic tensile deformation appears, and it evolves plastic failure in the posterior stage leading to the coarse fracture surface as shown in Figure 9A2. Compared to sample CIM, the fracture morphology of V_1_ presents two totally different regions, which is associated with crystalline morphology. It is clear that the fracture in the shear layer (Figure 9B1) presents brittle fracture due to the high content of shish-kebab. However, in the core regime, the whole region manifests a rough morphology as shown in Figure 9B2, resulting from the inferior resistivity capacity of spherulites and the high initial stress or temperature. As for the sample V_2_, shown in Figure 9C, it should be noted that this specimen after suffering SCG process was cut along the direction of the original notch by a blade to observe the surface morphology because it was not broken during the whole fracture process. The extremely smooth surface is shear layer (Figure 9C1), which is a signal of no defects formed in this layer. While plastic tensile deformation appears in the spherulites layer, indicated by the presence of plastic deformation observed in Figure 9C2, and plastic deformation induces a large content of microcracks in the spherulites layer. It could be known that the results concerning the sample V_2_ are in agreement with those manifested in Figure 8. That is, multiple microcracks form in the spherulites layer after the slow crack growth process is complete.

The failure process of sample with the special alternated structure has been discussed above, and such unique phenomenon has never been reported for virgin polymers in the previous research. As demonstrated in Figure 6 and Figure 8, a large number of microcracks are formed in the spherulites layer for sample V_2_, which presents extremely different fracture process (shown in Figure 10) in contrast to sample V_1_ and CIM. This is so-called dispersion damage mechanism, which usually occurs in fiber reinforced materials under the stress, resulting in the large consumption of energy during the fracture process. That is to say, the multiple microcracks can share the stress loaded on the sample with initial notch, so the crack propagation process would be suppressed significantly. It can be stated that local stress concentrations induced by microvoids could initiate yielding in the amorphous region firstly, this is followed by the fragmentation of crystalline lamellae and partial chains unfolding. Then, a fibrillar structure is generated when a cluster of uniaxially oriented molecules and intermediary voids are developed. Thus, the damage region grows into a crazing zone. After the craze is initiated, a crack slowly evolves when the fibrils fail under stress due to disentanglement and break-up of interlamellar tie molecules. The reasons for the formation of multiple microcracks in the spherulites of sample V_2_ may be summarized as follows: the interface adhesion strength between spherulites and shear layer is poor when the thermal annealing process is absent, leading to the easier formation of defects at the interface. So, many defects would occur under the conditions used in this work. These defects would propagate into microcracks, and microcracks are more easily formed in the spherulites layer resulting from the inferior capacity for resisting load. These factors cause the dispersion damage phenomenon for sample V_2_. However, some studies are still required for further accurately understanding this unique phenomenon.

## 4. Conclusions

In the current work, samples with extremely different microstructures were successfully prepared by using CIM and MFVIM, respectively. The crystalline structure and the slow crack growth behavior were investigated. The DSC and 2D-SAXS results indicated that the crystalline structure associated with the capacity for resisting slow crack propagation, such as the content of shish-kebab, the distance between the lamellae, and the perfection of the lamellae, scarcely varies for sample V_1_ and V_2_. Compared with sample CIM, the total fracture time for samples V_1_ and V_2_ are remarkably enhanced due to the large number of shish-kebabs originating from the extra imposed shear field during the packing stage. Interestingly, the slow crack growth behavior of sample V_2_ not only manifested the improvement of the complete fracture time, but also the unique crack propagation behavior, showing that multiple microcracks formed in the spherulites layer. This is so-called dispersion damage, forming a large number of microcracks, which results in the reduction of load for the initial notch, and can further enhance the service life of polymer products through the formation of multiple microcracks. The results of this work indicated the practical application prospect of a sample with a shear layer–spherulites layer alternated structure in the near future.

## Figures and Tables

**Figure 1 polymers-12-02746-f001:**
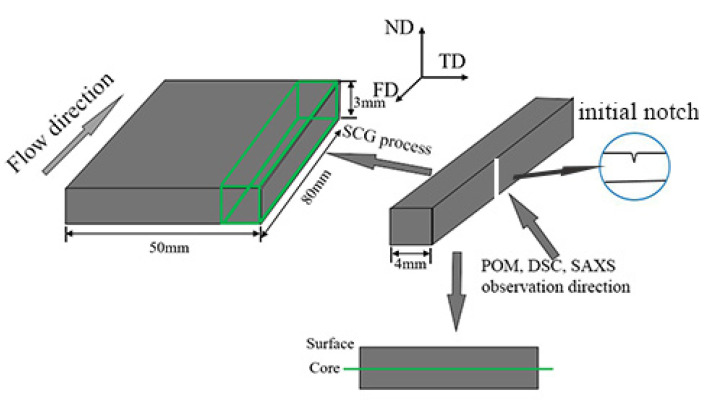
Schematic diagram of sample prepared for characterizations. FD, flow direction; TD, transverse direction; ND, normal direction.

**Figure 2 polymers-12-02746-f002:**
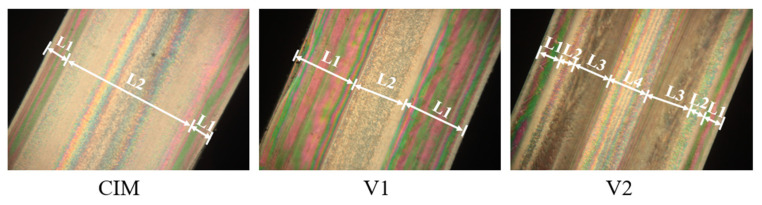
Polarized optical microscopy (POM) photographs of different specimens. L1 and L3 represent the shear layer, L2 and L4 are the spherulites layer.

**Figure 3 polymers-12-02746-f003:**
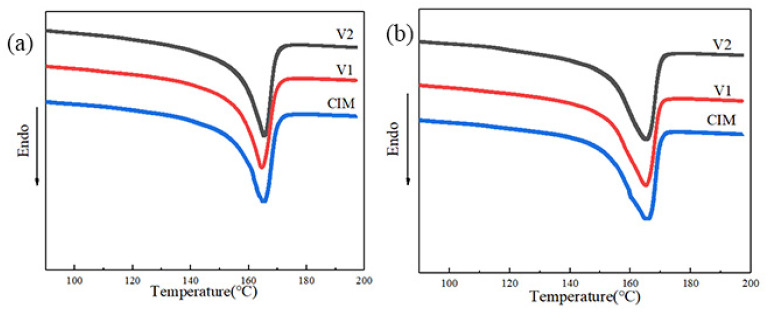
Differential scanning calorimeter (DSC) melting curves of different layers of samples, (**a**) shear layer, (**b**) spherulites layer.

**Figure 4 polymers-12-02746-f004:**
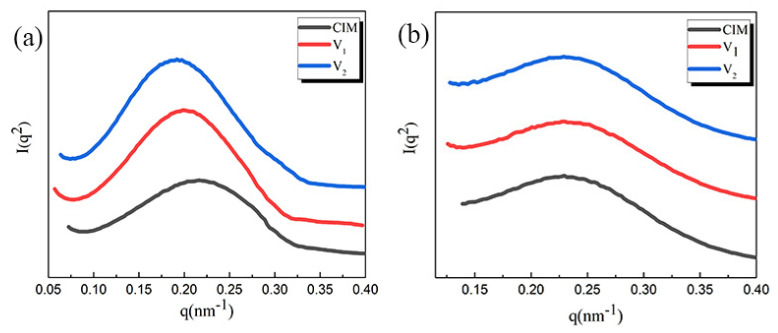
Corresponding intensity profiles of 1D-SAXS for different samples as a function of the scattering vector (q): (**a**) shear layer; (**b**) spherulites layer.

**Figure 5 polymers-12-02746-f005:**
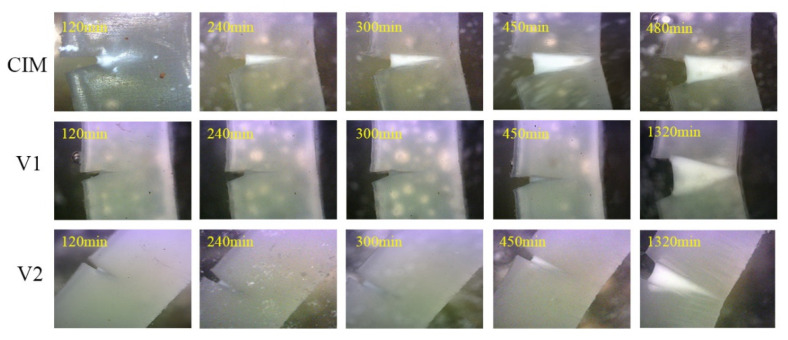
The patterns of the crack growth of specimens under a low initial stress for different times at 50 °C.

**Figure 6 polymers-12-02746-f006:**
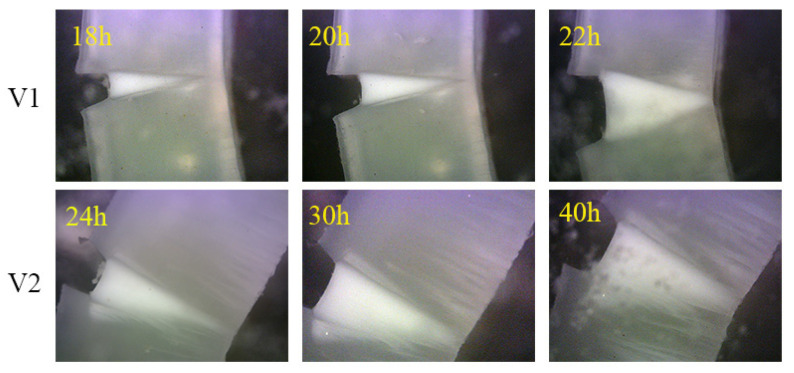
The patterns of crack propagation of sample at the later stage.

**Figure 7 polymers-12-02746-f007:**
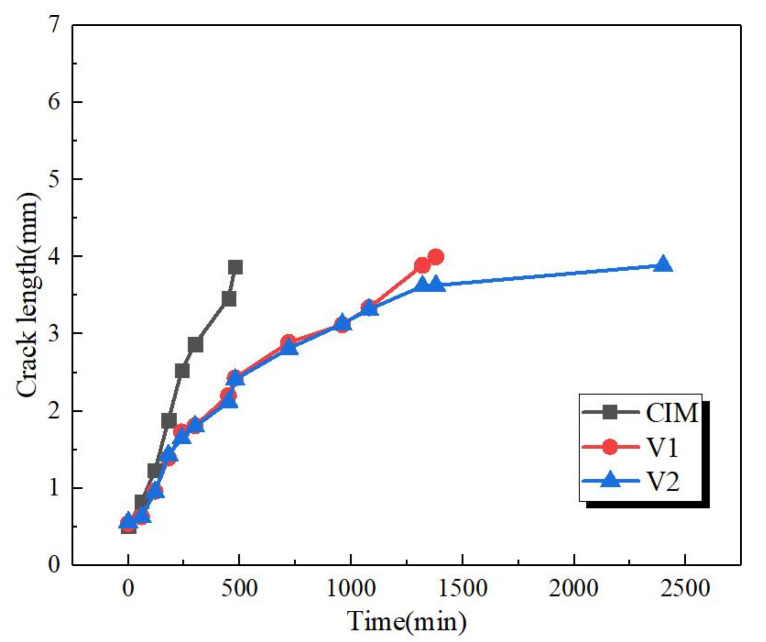
The crack length versus time of all the samples during the process of slow crack growth (SCG).

**Figure 8 polymers-12-02746-f008:**
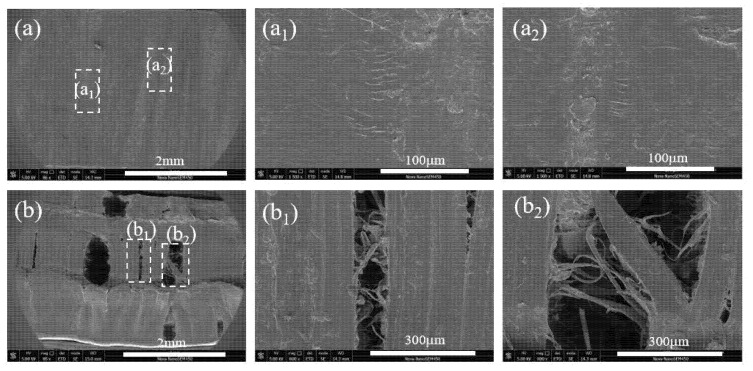
The SEM images of morphology of sample after suffering SCG process. (**a**): V_1_, (**b**): V_2_. (**a_1_**), (**a_2_**) and (**b_1_**), (**b_2_**) are the corresponding magnified images in (**a**) and (**b**) respectively. The observation direction of SEM is the transverse direction (TD) shown in Figure 1.

**Figure 9 polymers-12-02746-f009:**
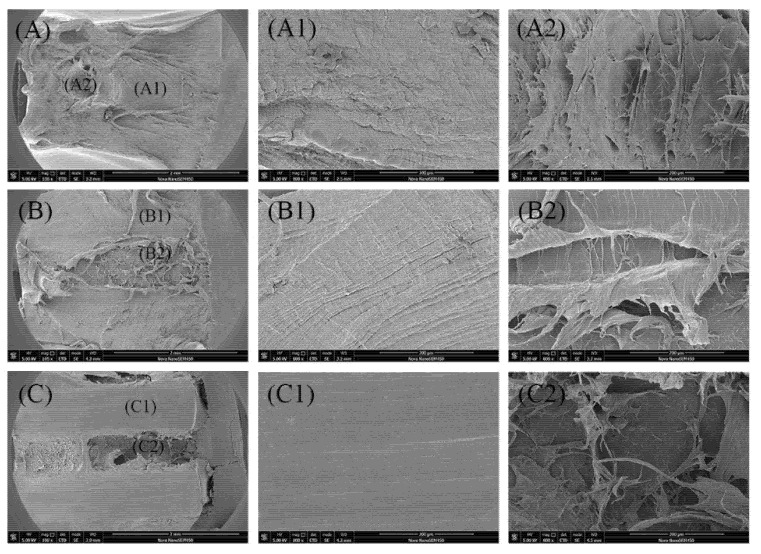
SEM images of the fracture morphology of specimens, the crack propagation direction is from right to left, (**A**) CIM; (**B**) V_1_; (C) V_2_. (**A1**), (**A2**), (**B1**), (**B2**) and (**C1**), (**C2**) are the corresponding magnified images in (**A**), (**B**) and (**C**), respectively.

**Figure 10 polymers-12-02746-f010:**
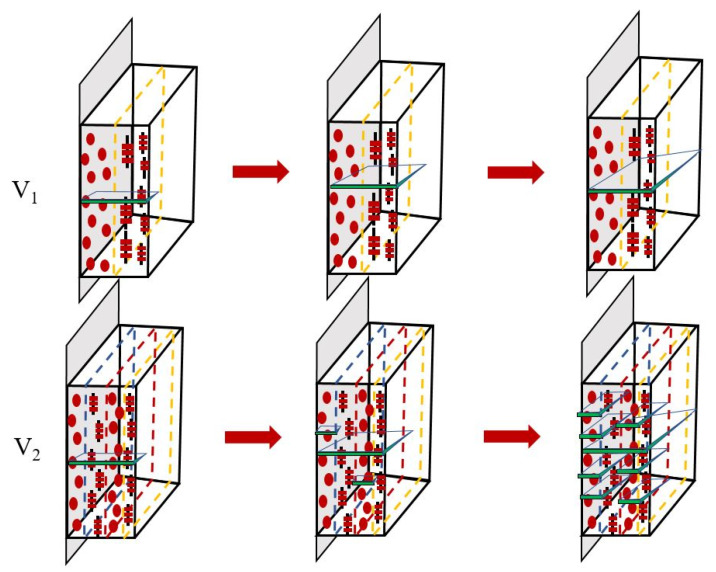
Schematic drawing of failure process of sample V_1_ and V_2_ under low initial load.

**Table 1 polymers-12-02746-t001:** Molding parameters of vibration injection molding process.

	CIM	V1	V2
Injection pressure (MPa)	40	40	40
Packing pressure (MPa)	30	30	30
Vibration pressure (MPa)		60	100
Interval time(s)		1/4/8	10/5

**Table 2 polymers-12-02746-t002:** Melting temperature and crystallinity of different layers of samples.

Sample	CIM	V_1_	V_2_
$T_{m}$ of shear layer (°C)	165.29	164.57	165.21
$T_{m}$ of spherulites layer (°C)	165.14	165.11	165.55
$X_{c}$ of shear layer (%)	44.4	44.6	45.3
$X_{c}$ of spherulites layer (%)	44.5	42.7	43.2

**Table 3 polymers-12-02746-t003:** Value of long period and crystalline size of different layers of samples.

Sample	CIM	V_1_	V_2_
$L_{p}$ of shish-kebab (nm)	28.98	31.67	31.81
$L_{p}$ of spherulites (nm)	27.36	27.29	27.41
$L_{c}$ of shish-kebab (nm)	12.87	14.12	14.41
$L_{c}$ of spherulites(nm)	12.18	11.65	11.84

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
