# Peer review of "Unique Slow Crack Growth Behavior of Isotactic Polypropylene: The Role of Shear Layer-Spherulites Layer Alternated Structure"

_polymers, 2020, doi:10.3390/polym12112746_

Round 1

Reviewer 1 Report

The authors have looked into iPP samples with alternating structures of shear and spherulites layer. They found that the alternated structures improved the resistivity of the polymer to slow crack growth. This paper is well written. There are some minor concerns that should be addressed by the authors.

1. Figures 5 should be followed by the text explaining it. The same applies to Figure 6. Currently, the figures appear together followed by a long text referring to both.

2. Can the authors comment on the effect of the alternated structure (shear layer-spherulite layer) on the mechanical properties of iPP?   3. Additionally, the document has minor typological errors that need to be corrected.  

Author Response

Dear reviewer,

Thank you for the comments which are very helpful for improving our paper. We have studied the comments carefully and have revised the manuscript accordingly.

The main corrections in the manuscript and the response to the reviewer’s comments are as following:

Reviewer #1:

The authors have looked into iPP samples with alternating structures of shear and spherulites layer. They found that the alternated structures improved the resistivity of the polymer to slow crack growth. This paper is well written. There are some minor concerns that should be addressed by the authors.

  1. Figures 5 should be followed by the text explaining it. The same applies to Figure 6. Currently, the figures appear together followed by a long text referring to both.

Response: Thanks for your comment. We have already made modifications in the corresponding locations

  1. Can the authors comment on the effect of the alternated structure (shear layer-spherulite layer) on the mechanical properties of iPP?  

Response: Thanks for your comment. The effect of the alternated structure (shear layer-spherulite layer) on the mechanical properties of iPP has been published in our previous papers.(F. Hou, D. Mi, M. Zhou, J. Zhang, The influences of a novel shear layer-spherulites layer alternated structure on the mechanical properties of injection-molded isotactic polypropylene,  Polymer, 122 (2017) 12-21. and M. Liu, R. Hong, X. Gu, Q. Fu, J. Zhang, Remarkably improved impact fracture toughness of isotactic polypropylene via combining the effects of shear layer-spherulites layer alternated structure and thermal annealing,  Industrial & Engineering Chemistry Research, 58 (32) (2019) 15069-15078.). Specifically, the impact strength for sample with alternated structure is nearly 30KJ/m2, which can further be enhanced (90KJ/m2) under the suitable annealing condition. As for tensile strength, it increased from 35MPa for CIM to 50MPa for sample with alternated structure due to the formation of shish-kebab.

  1. Additionally, the document has minor typological errors that need to be corrected.  

Response: Thanks for your careful work. We have tidied up the language as far as we can.

Thank you and best regards.

Yours sincerely,

Mingjin, Liu

Name: Jie Zhang

E-mail: zhangjie@scu.edu.cn

Reviewer 2 Report

The manuscript reports about an experimental campaign on the slow crack growth (SCG) behaviour of isotactict polypropilene (iPP). Three types of specimens were manufactured by using (i) conventional injection molding (CIM) and multiflow vibration injection molding (MFVIM) with (ii) increased shear layer thickness and (iii) shear layer-spherulites layer alternated structure. Results are compared and discussed for the three cases.

The presented research is interesting and in line with the scope of the journal.

However, from a Fracture Mechanics perspective, it would be useful to know further information about some quantitative measures characterising crack growth, e.g. the measured crack opening displacement (at the outer edge of the sample and/or at the initial crack tip) and crack length, as functions of the corresponding applied load or displacement. If such information is available, it should be reported in the manuscript.

Besides, minor changes are necessary. Detailed comments follow.

1) Samples Preparation
It is mentioned that "0.5-0.6mm notch was made in the specimens". Authors should clarify:
- where exactly were the notches made?
- which tool was used to cut the specimens?
- were the crack tips sharp or blunted? (this is relevant to apply Fracture Mechanics concepts)
- how were the initial crack lengths measured? (even small differences in the initial crack length may produce different behaviours)
A schematic figure of the notched specimens would help.

2) Testing Procedure
Authors state: "The initial stress we selected for sample CIM was 5.9 MPa, and it was 7.2MPa for both sample V1 and V2". However some pieces of information are missing:
- which testing machine was used?
- which was the testing setup?
- was the stress level kept fixed during the tests? or was a fixed displacement imposed?
A schematic figure of the testing setup would help.

3) English Language
- acronyms should be defined at their first occurrence: in particular, check "iPP" in the Abstract and at line 67; "PE" at line 43;
- some long sentences should be made shorter for clarity: for instance, in the Abstract "Here, we manufactured ... before.", etc.
- typos should be fixed: at lines 57 and 84, "multflow" should be "multiflow"; at line 145, "detialed" should be "detailed"; at line 146, "orinetation" should be "orientation";
- check the use of some words: e.g., "literatures" appearing several times should be replaced by "literature" or "researches"; at line 177, "semblable" should be "similar" (?);
- the last sentence of the Abstract ("This work gave ... application prospects.") is too general. Either be more specific or remove it.

Author Response

Dear reviewer,

Thank you for the comments which are very helpful for improving our paper. We have studied the comments carefully and have revised the manuscript accordingly.

The main corrections in the manuscript and the response to the reviewer’s comments are as following:

Reviewer #2

The manuscript reports about an experimental campaign on the slow crack growth (SCG) behavior of isotactic polypropylene (iPP). Three types of specimens were manufactured by using (i) conventional injection molding (CIM) and multiflow vibration injection molding (MFVIM) with (ii) increased shear layer thickness and (iii) shear layer-spherulites layer alternated structure. Results are compared and discussed for the three cases.

The presented research is interesting and in line with the scope of the journal.

However, from a Fracture Mechanics perspective, it would be useful to know further information about some quantitative measures characterising crack growth, e.g. the measured crack opening displacement (at the outer edge of the sample and/or at the initial crack tip) and crack length, as functions of the corresponding applied load or displacement. If such information is available, it should be reported in the manuscript.

Response: Thanks for your comment. There is no doubt that from a Fracture Mechanics perspective, it would be useful to know further information about some quantitative measures characterising crack growth. Here, we add a diagram presenting the crack length versus time of all the samples during the process of SCG(Fig 7).

Besides, minor changes are necessary. Detailed comments follow.

1) Samples Preparation
It is mentioned that "0.5-0.6mm notch was made in the specimens". Authors should clarify:
- where exactly were the notches made?
- which tool was used to cut the specimens?
- were the crack tips sharp or blunted? (this is relevant to apply Fracture Mechanics concepts)
- how were the initial crack lengths measured? (even small differences in the initial crack length may produce different behaviors)
A schematic figure of the notched specimens would help.

Response: Thanks for your comment. We made a notch in the middle location for all the samples(shown in Figure 1) by a blade. The crack tips were sharp and the initial crack lengths were measured by vernier caliper. It should be noted that we tested three times for each kind of sample to avoid small differences in the initial crack length may produce different behaviors. A schematic figure of the notched specimens has been presented in Figure 1.

2) Testing Procedure
Authors state: "The initial stress we selected for sample CIM was 5.9 MPa, and it was 7.2MPa for both sample V1 and V2". However some pieces of information are missing:
- which testing machine was used?
- which was the testing setup?
- was the stress level kept fixed during the tests? or was a fixed displacement imposed?
A schematic figure of the testing setup would help.

Response: Thanks for your comment. We used spring scale to correct the real initial stress loading on the samples. The stress was not kept fixed during the tests, but the variation amplitude of stress should be the same under the same displacement condition.And we selected the same stress for this two sampes to focus on the differce between sampleV1 and V2. A schematic figure of the notched specimens could be found in the previous work. (Y. Pan, X. Gao, Z. Wang, J. Lei, Z. Li, K. Shen, Effect of different morphologies on slow crack growth of high-density polyethylene, Rsc Advances, 5 (36) (2015) 28191-28202.)

3) English Language
- acronyms should be defined at their first occurrence: in particular, check "iPP" in the Abstract and at line 67; "PE" at line 43;
- some long sentences should be made shorter for clarity: for instance, in the Abstract "Here, we manufactured ... before.", etc.
- typos should be fixed: at lines 57 and 84, "multflow" should be "multiflow"; at line 145, "detialed" should be "detailed"; at line 146, "orinetation" should be "orientation";
- check the use of some words: e.g., "literatures" appearing several times should be replaced by "literature" or "researches"; at line 177, "semblable" should be "similar" (?);
- the last sentence of the Abstract ("This work gave ... application prospects.") is too general. Either be more specific or remove it.

Response: Thanks for your careful work. We have tidied up the language as far as we can.

Thank you and best regards.

Yours sincerely,

Mingjin, Liu

Name: Jie Zhang

E-mail: zhangjie@scu.edu.cn
